# Cerebral Oxidative Stress in Early Alzheimer’s Disease Evaluated by ^64^Cu-ATSM PET/MRI: A Preliminary Study

**DOI:** 10.3390/antiox11051022

**Published:** 2022-05-22

**Authors:** Hidehiko Okazawa, Masamichi Ikawa, Tetsuya Tsujikawa, Tetsuya Mori, Akira Makino, Yasushi Kiyono, Yasunari Nakamoto, Hirotaka Kosaka, Makoto Yoneda

**Affiliations:** 1Biomedical Imaging Research Center, University of Fukui, Fukui 910-1193, Japan; iqw@u-fukui.ac.jp (M.I.); awaji@u-fukui.ac.jp (T.T.); morit@u-fukui.ac.jp (T.M.); amakino@u-fukui.ac.jp (A.M.); ykiyono@u-fukui.ac.jp (Y.K.); myoneda@fpu.ac.jp (M.Y.); 2Second Department of Internal Medicine, Faculty of Medical Sciences, University of Fukui, Fukui 910-1193, Japan; ynakamot@u-fukui.ac.jp; 3Department of Neuropsychiatry, Faculty of Medical Sciences, University of Fukui, Fukui 910-1193, Japan; hirotaka@u-fukui.ac.jp; 4Faculty of Nursing and Social Welfare Science, Fukui Prefectural University, Fukui 910-1195, Japan

**Keywords:** oxidative stress, Alzheimer’s disease, ^64^Cu-ATSM, ^11^C-PiB, PET/MRI

## Abstract

Oxidative stress imaging using diacetyl-bis (*N*^4^-methylthiosemicarbazone) (Cu-ATSM) was applied to the evaluation of patients with early Alzheimer’s disease (eAD). Ten eAD patients (72 ± 9 years) and 10 age-matched healthy controls (HCs) (73 ± 9 years) participated in this study. They underwent dynamic PET/MRI using ^11^C-PiB and ^64^Cu-ATSM with multiple MRI sequences. To evaluate cerebral oxidative stress, three parameters of ^64^Cu-ATSM PET were compared: standardized uptake value (SUV), tracer influx rate (K_in_), and a rate constant k_3_. The input functions were estimated by the image-derived input function method. The relative differences were analyzed by statistical parametric mapping (SPM) using SUV and K_in_ images. All eAD patients had positive and HC subjects had negative PiB accumulation, and MMSE scores were significantly different between them. The ^64^Cu-ATSM accumulation tended to be higher in eAD than in HCs for both SUV and K_in_. When comparing absolute values, eAD patients had a greater K_in_ in the posterior cingulate cortex and a greater k_3_ in the hippocampus compared with lobar cortical values of HCs. In SPM analysis, eAD had an increased left operculum and decreased bilateral hippocampus and anterior cingulate cortex compared to HCs. ^64^Cu-ATSM PET/MRI and tracer kinetic analysis elucidated cerebral oxidative stress in the eAD patients, particularly in the cingulate cortex and hippocampus.

## 1. Introduction

Oxidative stress is induced by various causes and mechanisms, including mitochondrial dysfunction, inflammatory changes, and hypoxic changes. It is considered one of the major molecular mechanisms of neuronal impairment and degeneration in various neurodegenerative diseases including parkinsonian syndromes, motor neuron diseases such as amyotrophic lateral sclerosis (ALS), and various forms of dementia [1,2,3]. Postmortem studies have revealed pathological and biochemical evidence of oxidative damage to the brain in these diseases [2,3,4]. Many previous papers have pointed out that oxidative stress plays a major role in the degenerative changes of the brain and spinal cord in neuronal disorders. However, it has been difficult to evaluate the extent of oxidative damage during the neurodegenerative process in the in vivo brains of patients. To elucidate the oxidative stress involved in pathologic changes in the brain in vivo, we performed positron emission tomography (PET) studies on patients with mitochondrial myopathy, encephalopathy, lactic acidosis, and stroke-like episodes (MELAS) syndrome, Parkinson’s disease (PD), and ALS using radioactive copper labeled diacetyl-bis(*N*^4^-methylthiosemicarbazone) (Cu-ATSM) [5,6,7]. The mechanism of intracellular retention of radioactive copper is based on the reduction of divalent copper [Cu(II)] to monovalent copper [Cu(I)] in over-reduction microenvironments with excessive electrons, such as hypoxia and oxidative stress. The reduced Cu(I) dissociates from the ATSM complex and eventually becomes irreversibly trapped in the cell [8,9,10,11].

In Alzheimer’s disease (AD), the most common neurodegenerative disorder that causes dementia, oxidative stress in the brain is also considered to be induced by reactive oxygen species (ROS) generated by the activation of microglial and glial responses to amyloid aggregation and/or neurofibril tangles [12,13,14,15,16,17]. Although there are some AD patients whose disease may not necessarily be pathologically caused by amyloid aggregation or neurofibril tangles, the latter are closely related to neuroinflammatory changes and the clinical severity of dementia [18,19,20]. Imaging of the translocator protein (TSPO) and monoamine oxidase B (MAO-B) has been developed to observe microglial and glial activation in neurodegenerative regions [21,22]. Although these probes cannot directly reflect ROS or oxidative stress, which are considered to be the direct causes of neurodegeneration, Cu-ATSM PET can assess the degree of oxidative stress in the brains of AD patients, compared with healthy controls (HCs) [5,6,7].

In the present study, ^64^Cu-ATSM was used to evaluate the degree of oxidative stress in the brains of AD patients, in addition to PET/MRI studied with amyloid imaging. We sought to investigate pathophysiologic changes in the eAD brain in terms of oxidative stress and to assess whether an option of treatment that reduces oxidative stress could be promising for the prevention of neuronal degeneration. If the distribution of ^64^Cu-ATSM is similar to the region responsible for eAD, then reducing oxidative stress in the region would be a potential therapeutic target. The AD group was determined by cognitive decline and positive accumulation of ^11^C-labeled Pittsburg Compound B (PiB), a representative amyloid imaging probe, in the cerebral cortex.

## 2. Materials and Methods

### 2.1. Subjects

Ten patients with early AD (eAD ) and 10 age-matched HCs participated in this study (Table 1). All subjects were diagnosed at a consensus conference. All patients with eAD met National Institute for Aging-Alzheimer’s Association (NIA/AA) criteria for probable AD dementia [23]. They underwent a medical and neurological workup with a neuropsychological battery that included the Mini-Mental State Examination (MMSE) and Clinical Dementia Rating (CDR). All patients participating in this study were assessed for positive ^11^C-PiB accumulation in the cerebral cortices. All subjects underwent PET/MRI using ^11^C-PiB and ^64^Cu-ATSM on separate days at the Biomedical Imaging Research Center, University of Fukui. The study was approved by the Ethics Committee of the University of Fukui, Faculty of Medical Sciences (study protocol # 20160124), based on its guidelines (Ethical Guidelines for Medical Science Research with Humans) as well as the Helsinki Declaration of 1975 (revised in 1983). Written informed consent was obtained from each subject.

### 2.2. PET/MRI Image Acquisition

For PET and MRI data acquisition using ^11^C-PiB and ^64^Cu-ATSM, a whole-body PET/MRI scanner (Signa PET/MR, ver. 26, GE Healthcare, Milwaukee, WI, USA) was used with a standard head coil (8-channnel HD Brain, GE Healthcare) [24,25]. The scanner permits PET data acquisition of 89 image slices in 3D mode, with a slice thickness of 2.76 mm [26]. The performance test showed the intrinsic resolution of PET images to be 4.2–4.3 mm full width at half maximum (FWHM) in the transaxial direction. The PET/MRI scanner was calibrated beforehand with a dose-calibrator (CRC-12, Capintec Inc., Florham Park, NJ, USA) using a pool phantom and ^18^F-solution, according to the scanner manufacturer’s guidelines [24].

A list-mode 3D PET scan was started using the time-of-flight acquisition mode at the time of bolus injection of the tracer from the antecubital vein. The scanning time and injection dose were 70 min and 700–750 MBq for ^11^C-PiB, and 40 min and 370 MBq for ^64^Cu-ATSM [27]. To confirm the accuracy of the blood radioactivity curve, a couple of 0.5 mL venous blood samples were obtained manually at the end of the PET scan and the concentration of radioactivity was measured with a gamma-well counter (ARC-8000, Hitachi-Aloka, Tokyo, Japan). A portion of the sampled blood (0.1 mL) was used for measurement of the octanol extraction to estimate the undissociated ratio of ^64^Cu-ATSM in the blood [11,28]. Multiple MRI sequences were acquired during the PET scan, including a 3D-T1 weighted image (WI), T2WI, FLAIR, MR angiography, and resting state functional MRI (rs-fMRI), as well as PET attenuation correction (AC) image data [25]. For MR-AC data, 3D radial MR acquisition was performed in the axial direction using the zero-echo time (ZTE) method with the following parameters: field of view (FOV), 264 mm; matrix, 110 × 110 × 116; voxel size, 2.4 × 2.4 × 2.4 mm^3^; flip angle, 0.8°; number of excitations, 4; bandwidth, ± 62.5 kHz; and acquisition time, 41 s [25]. Details of the ZTE-AC method are described elsewhere [29,30]. High-resolution 3D-T1WI anatomical MRI was collected using the following parameters [27,31]: repetition time = 8.5 ms; echo time = 3.2 ms; flip angle = 12°; FOV = 256 mm; 256 × 256 matrix; 136 slices; voxel dimension = 1.0 × 1.0 × 1.0 mm^3^.

Dynamic PET images were reconstructed from PET and MR-AC data using the 3D ordered subset expectation maximization (OSEM) method and point spread function modeling algorithm in 36 frames (12 × 5 s, 6 × 10 s, 3 × 20 s, 4 × 30 s, 5 × 60 s, 4 × 5 min, and 1 × 10 min) for ^64^Cu-ATSM and 39 frames (the same frames plus additional 3 × 10 min frames) for ^11^C-PiB [27,31]. The following OSEM parameter set was applied to the reconstruction of both PET images: subset, 28; iteration, 3; transaxial post-gaussian filter cutoff, 3 mm in 256 mm FOV, and 2 × 2 mm^2^ pixel size [27,31]. The decay of radioactivity in dynamic PET data was corrected to the starting point of each scan.

### 2.3. Calculation of Parametric Images

In order to evaluate cortical ^11^C-PiB accumulation, standardized uptake value (SUV) images were calculated from the average image 50–70 min after injection, i.e., the mean of the last two frames of the dynamic ^11^C-PiB PET data, using the body weight and injection dose of each subject. For evaluation of oxidative stress in the brain, three parameters of ^64^Cu-ATSM PET were calculated: (1) the SUV calculated from average images of 15–40 min time frames, (2) the tracer influx rate (K_in_) obtained by the Patlak plot method [32], and (3) the k_3_ value estimated by the non-linear least squares fitting (NLS) method. The input function for the parameter calculation was estimated by the image-derived input function (IDIF) method. Details of the IDIF method are described elsewhere [24]. In order to calculate the K_in_ and k_3_, the arterial input curve of the IDIF was corrected by the ^64^Cu fraction, remaining undissociated from ATSM in the blood (C_p_ [mL g^−1^]) [11,33]. Based on the retention mechanism of ^64^Cu-ATSM, the K_in_ (mL min^−1^ g^−1^), defined as K_1_∙k_3_/(k_2_+k_3_) or k_3_ (min^−1^), is considered to reflect the degree of regional oxidative stress, where K_1_–k_3_ are rate constants of the tracer (Figure 1). The Patlak plot method was applied to obtain the K_in_ image by a pixel-by-pixel slope calculation with the following equation:C_t_(t)/C_p_ (t) = K_in_∙ [Int C_p_ dt]/C_p_ (t)
where C_t_ is the brain tissue activity, and [Int C_p_ dt] means the integral of the plasma input [32].

### 2.4. Image Analysis

To obtain regional parametric values from ^64^Cu-ATSM images of the brain, multiple regions of interest (ROIs) were drawn on individual 3D-T1WI MRI images using PMOD software (version 3.9; PMOD Technologies Ltd., Zurich, Switzerland). An example of ROIs for a single subject is given in Figure 2. The MRI and PET brain coordinates were in exactly the same space because the images were acquired simultaneously by the PET/MRI scanner. The 3D-T1WI MRI was resliced to correspond with the slice levels of the PET image. To obtain SUV and K_in_ values for each cortical region, three image slices were used to set ROIs in the bilateral hemispheres. The k_3_ values were calculated from the individual C_t_ obtained from dynamic ^64^Cu-ATSM PET data using the same ROIs, and C_p_ by IDIF. The NLS method was applied for regional k_3_ estimation. Statistical parametric mapping (SPM12; Available online. https://www.fil.ion.ucl.ac.uk/spm/ (accessed on 26 April 2022), Wellcome Trust Centre for Neuroimaging, London, UK) was used to observe the regional differences between SUV and K_in_ images, as well as differences between groups for each image.

### 2.5. Statistical Analysis

Data are presented as the mean ± standard deviation (SD). All statistical analyses were performed using SPSS statistics (version 23; IBM Corporation, Armonk, NY, USA), and *p* < 0.05 was considered statistically significant. Group differences in the demographic characteristics of the participants were assessed using a two-tailed Mann–Whitney *U* test. Regional parametric values of the two groups were compared using an analysis of variance (ANOVA) with a post hoc Fisher’s *F*-test to analyze differences. Six brain regions of the frontal, temporal, parietal, and occipital lobes, hippocampus, and posterior cingulate cortex (PCC) were compared.

Regional differences between the images of the two groups were analyzed by SPM12. PET images were anatomically normalized to the Montreal Neurological Institute (MNI) space with a voxel size of 2 × 2 × 2 mm^3^ using individual 3D-T1WI images and the template provided by SPM12. After the normalized images were spatially smoothed with a 10 mm Gaussian filter for group comparison, each image pertaining to z values was entered into SPM12 and a two-sample *t*-test was performed. We applied a statistical threshold of *p* < 0.001 uncorrected at the voxel level and *k* > 100 with *p* < 0.05 uncorrected at the cluster-level for multiple comparisons across the whole brain. Statistical mapping analysis was also performed for rs-fMRI data. Details of the group analysis are described in a previous report on multimodality data analysis [27].

## 3. Results

There were no significant differences in the distributions of age, disease duration, and education between patients with eAD and the HCs. The mean MMSE score was significantly lower in the eAD group (23.7 ± 2.6) than in the HC group (29.2 ± 0.9, *p* < 0.0001) (Table 1). The mean CDR and CDR sum of boxes (CDR-SB) scores were significantly greater in eAD patients than in HCs (*p* < 0.0001 for both). As confirmed by the ^11^C-PiB PET images, all patients were found to have positive cortical accumulation of PiB and all HC subjects had negative accumulation. SUV and K_in_ images of representative cases are presented in Figure 3. The distribution of both parametric values was similar, with cortical accumulation tending to be higher in eAD than in HCs, especially in the PCC and hippocampus.

A representative whole arterial blood curve obtained from the IDIF and a plasma curve (C_p_) corrected for ^64^Cu dissociation are shown in Figure 4A. The insert of this graph shows the undissociated rate of ^64^Cu-ATSM as a function of time. The radioactivity concentrations of manual blood sampling agreed well with those estimated by the IDIF method. The ratio of octanol extraction to whole blood radioactivity was also consistent with the undissociated rate curve of ^64^Cu-ATSM. Figure 4B shows the Patlak plot of a cortical region for the same case. A good linear regression (*r*^2^ = 0.994) provided a slope representing the K_in_ value for this region. Figure 4C is the result of NLS fitting of the same region to obtain the regional k_3_. The radioactivity curve of brain tissue fitted well to the two-tissue compartment three-parameter model.

Table 2 shows regional values of the three parameters obtained from the manual ROI method. All cortical parameter values of SUV, K_in_ and k_3_ for ^64^Cu-ATSM tended to be greater in eAD compared with HCs. Although there were no regional differences between HCs and eAD in SUV, the PCC of eAD patients showed a significantly greater K_in_ compared with the frontal and parietal lobes of HC subjects (*p* < 0.05, Figure 5A). The hippocampal k_3_ of eAD patients was significantly greater than that of the major lobes and PCC in HC subjects (*p* < 0.05, Figure 5B). The K_1_ in the hippocampus was significantly lower than that in the major lobes and PCC within each group (*p* < 0.01, Figure 5C). In SPM analysis, there was no difference between the SUV and K_in_ in either group. When comparing regional differences of SUV and K_in_ between eAD and HCs, eAD showed reductions in the bilateral hippocampus (peaks at [20, -12, -16] and [-18, -4, -16]) and anterior cingulate cortex (ACC) (peak at [0, 28, 16]), and a significant increase in the left central operculum (peak at [-34, 6, 12], cluster-level P_uncorr_ < 0.05) (Figure 6). The difference in the ACC was also observed in the rs-fMRI analysis (peak at [0, 27, 24], *p* < 0.001, k > 100).

## 4. Discussion

Oxidative stress induced by ROS generation in the brain is considered to be one of the important causes of neuronal degeneration in eAD patients. In the present study, a tendency for a global increase in accumulation of ^64^Cu-ATSM in the cerebral cortices of eAD was observed, and the kinetic parameters, reflecting the over-reduced state, indicated oxidative stress in specific brain regions. K_in_ and k_3_ values were significantly increased in the PCC and hippocampus of the eAD brain, i.e., the specific regions where amyloid-β and tau proteins gradually accumulate depending on the stage of AD [16,17,34]. This result indicates that global and regional increases in oxidative stress are probably mainly caused by mitochondrial dysfunction in the brains of eAD patients in association with pathological protein deposition. In contrast, SPM analysis showed a relative increase in the left central operculum and decreases in the bilateral hippocampus and ACC on both SUV and K_in_ images. These regional differences between ROI analysis and SPM may be caused by the difference in the methods of regional evaluation by absolute values and statistical mapping after global normalization.

A number of postmortem and biochemical studies have indicated that oxidative stress plays a major role in neuronal degeneration in AD. Multiple pathological studies showed that the temporal lobes of eAD patients have elevated levels of oxidation products, such as protein carbonyls and 8-hydroxyguanosine, compared with those of healthy individuals [12,35]. Recent studies showed that tau and microglial activation are closely related to neuronal degeneration, especially in the medial temporal regions, in the stages of MCI to eAD [21,36,37], although there are some AD patients whose disease may not necessarily be pathologically caused by amyloid aggregation or neurofibril tangles [18,19,20]. However, neurofibril tangles are usually closely related to dystrophic neuritis and the clinical severity of dementia. The present study showed a tendency for increased ^64^Cu-ATSM accumulation in all cerebral cortex regions, as well as regional changes in the hippocampus and cingulate cortices. These results were consistent with cortical ^11^C-PiB accumulation reflecting amyloid deposition, and regional pathologic changes in the previous studies [16,17,21]. Although there are several causes of oxidative stress induction other than mitochondrial dysfunction, including hypoxia and inflammatory changes, mitochondria and their deficiencies are the largest source of ROS [38]. There is no clear evidence of a relationship between the global increase in ^64^Cu-ATSM accumulation and the pathogenesis eAD; however, one of the potential reasons may be changes in the microenvironment induced by the impaired clearance function of waste products in the brain [39]. To evaluate regional and dynamic changes of oxidative status in living patients, ^64^Cu-ATSM PET with kinetic analysis seems to be an ideal approach. The imaging method may be able to be applied to other neurodegenerative diseases which are relevant with amyloid aggregation and neurofibril tangles [40,41,42,43].

For kinetic analysis and calculation of parametric images, the IDIF method was used in the present study. This method is very useful for calculating absolute values of kinetic parameters in the brain using a PET/MRI scanner [24,27,31]. However, metabolite correction is usually required, except for in ^15^O-water studies, and collecting venous blood to measure metabolite ratios is recommended as a non-invasive method when using the IDIF [33,44]. We collected venous blood samples at the end of the PET scan to confirm the radioactivity of the whole blood as well as the undissociated ratio of ^64^Cu-ATSM in the blood. The radioactivity concentration obtained by the IDIF curve was consistent with that of the sampled blood, and the undissociated ratio was also very close to the ratio calculated from the equation in the previous report [11]. Therefore, the input function estimated by our method is considered to be appropriate for kinetic analysis and image calculation. If the PET tracer can be analyzed with a two-tissue compartment four-parameter model, the reference tissue method can be applied for kinetic analysis, but in the image calculation of ^64^Cu-ATSM, a two-tissue compartment three-parameter model, the Patlak plot method should be applied based on the retention mechanism (Figure 1). For evaluation of mitochondrial function, the rate constant k_3_ estimated by the NLS method may be the most appropriate according to the retention mechanism of ^64^Cu-ATSM.

The regional differences between the HC and eAD in absolute values of K_in_ and k_3_ were not the same according to the ROI analysis. The real difference in oxidative stress may only be assessed by the absolute values of kinetic rate constants, because the SUV and K_in_ are considered to be affected by K_1_. The decrease in the K_in_ in the hippocampus revealed by SPM analysis may have been caused by relative local changes in eAD, provably due to neuronal dysfunction (Figure 5), whereas the k_3_ value in this region was significantly elevated. The remarkable k_3_ increase in the hippocampus is noteworthy because the K_1_ of the region was markedly lower than in other cortical regions. The relative hippocampal K_in_ reduction in the eAD brain is consistent with the results of ROI analysis, which showed a relatively small K_in_ increase in the hippocampus compared with other cerebral regions (Figure 5A), probably due to the lower regional K_1_ (Figure 5C). These results suggest the importance of the k_3_ in the evaluation of oxidative stress using ^64^Cu-ATSM PET, especially in regions where cerebral blood flow (CBF) is altered. In order to assess regional changes in oxidative stress due to mitochondrial dysfunction, regional differences in CBF between groups may need to be considered.

We previously measured regional changes in oxidative stress in patients with Parkinson’s disease and ALS using ^62^Cu-ATSM PET, showing the possibility of oxidative stress-induced neuronal degeneration in the responsible area [6,7]. In the present study, the same PET probe labeled by ^64^Cu was applied to eAD patients using a PET/MRI scanner. The quantitative parameters were calculated and compared using an arterial input function obtained by the IDIF method, one of the advantages of PET/MRI. The K_in_ and SUV images were similar, and SPM analysis of each group showed no relative difference between the two images, indicating that relative regional changes could be analyzed using SUV and statistical mapping methods to compare tracer influx rates. In contrast, ROI analysis did not show significant differences in regional SUVs, suggesting that in order to compare absolute regional changes of ^64^Cu-ATSM accumulation, it may be necessary to normalize the SUVs using a reference region, as used in the previous studies [6,7].

The SPM analysis using SUV and K_in_ images showed a significant difference in the ACC, although regional changes may be relatively small, e.g., as seen in the hippocampus. The difference in the ACC was also observed in the rs-fMRI analysis, although the change was a functional decrease [27]. Functional reduction without oxidative stress in the ACC may have caused the decrease in CBF and ^64^Cu-ATSM accumulation. The ACC plays a major role in working memory that requires attentional shifting, as well as in positive motivation and cognitive conflict [45,46], all of which are associated with AD symptoms. Similarly, a functional connectivity study showed disruption of the cingulo-opercular network in AD patients, especially those with apathy [47], suggesting that functional changes in these regions, including the operculum, may have altered the redox states in the brains of AD patients. Although statistical mapping analyses using CBF and glucose metabolism in eAD patients have not reported significant differences in these areas, only impairment of neuronal connectivity and mitochondrial function without pathologic changes may have occurred in the early stages of the disease.

This study has some limitations including the limited number of subjects studied. Since this is the first study to analyze the K_in_ and k_3_ of Cu-ATSM in the brain, we wanted to assess the feasibility of a quantitative method that reflects tracer reduction. As shown in Figure 1, the tracer kinetics and trapping mechanism have already been confirmed [8]. Therefore, the NLS fitting results indicate that the two-tissue compartment three-parameter model is suitable for analysis in Cu-ATSM studies. Further studies using this method are expected to confirm our findings.

## 5. Conclusions

^64^Cu-ATSM PET/MRI revealed oxidative stress in the brain of eAD patients, especially in the cingulate cortex and hippocampus. These findings indicate the relationship between oxidative stress and pathological protein deposition in eAD. The results of oxidative stress in the eAD brain may indicate the possibility of a new treatment method reducing oxidative stress for the prevention of neuronal degeneration. Further studies with a combination of ^64^Cu-ATSM and tau PET/MRI will reveal the effects of oxidative stress on the progress of neurodegeneration in AD patients.

## Figures and Tables

**Figure 1 antioxidants-11-01022-f001:**
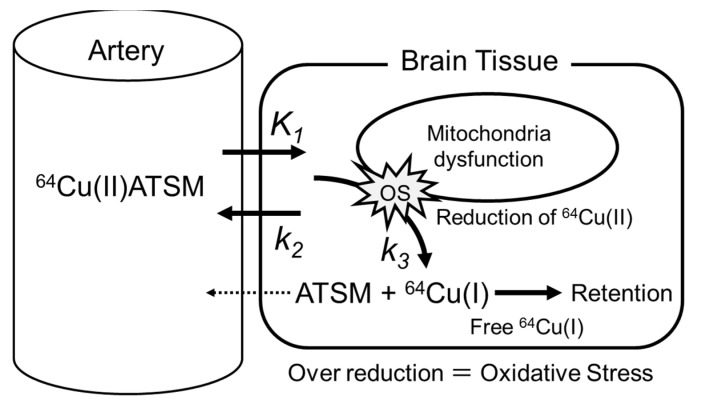
Kinetic model of ^64^Cu-ATSM. ^64^Cu(II) is reduced to ^64^Cu(I) and irreversibly dissociated from ATSM chelate. Therefore, the 1-tissue compartment 3-parameter model represents the tracer kinetics well.

**Figure 2 antioxidants-11-01022-f002:**
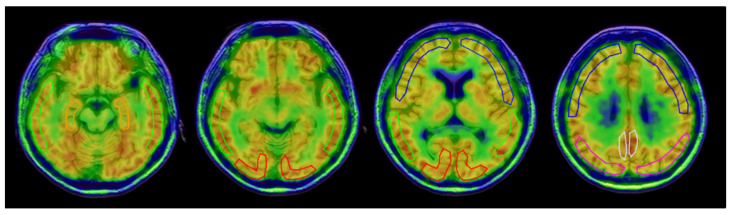
An example of ROIs for frontal (blue), temporal (green), parietal (magenta), and occipital (red) lobes, hippocampus (orange) and posterior cingulate cortex (gray) drawn in a single subject.

**Figure 3 antioxidants-11-01022-f003:**
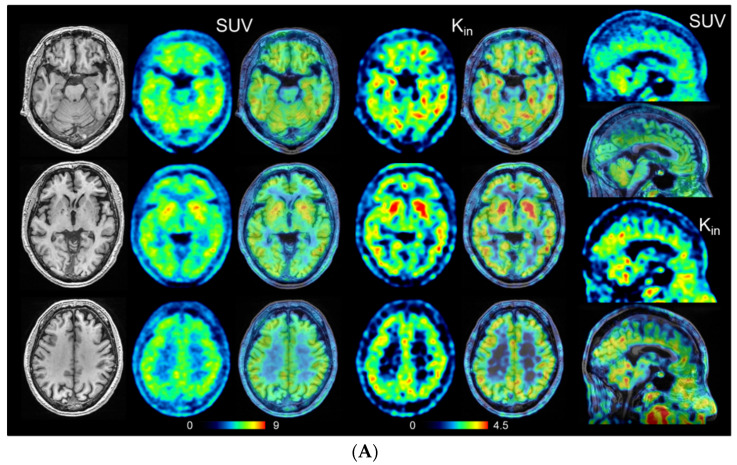
Representative images of HCs (**A**) and eAD (**B**) for SUV and K_in_ (mL/min/100 g) PET image (left) and fusion image (right). The right end column shows sagittal view of SUV (top two rows) and K_in_ (bottom two rows). The left end column shows T1WI-MRI of identical slice location. Note accumulations of the PCC and hippocampus are elevated in eAD patients compared with HCs.

**Figure 4 antioxidants-11-01022-f004:**
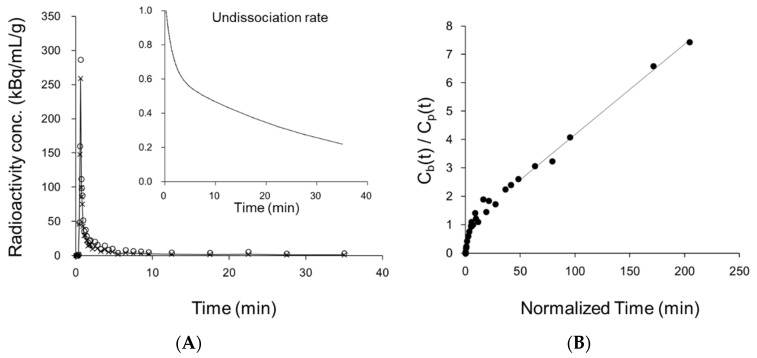
(**A**) The whole arterial blood radioactivity concentration was estimated by the IDIF method (circle), which was then corrected to the real input function (x and solid line) by the undissociated rate of ^64^Cu-ATSM (insert) [11]. (**B**) The Patlak plot calculated from the corrected IDIF (C_p_) and cerebral tissue counts (C_t_) shows good linearity in the later phase of ^64^Cu-ATSM PET scan (slope: 0.032, *r*^2^ = 0.994). The slope represents K_in_ value. (**C**) Representative NLS fitting based on two-tissue compartment three-parameter model. The fitting provides an estimate of k_3_.

**Figure 5 antioxidants-11-01022-f005:**
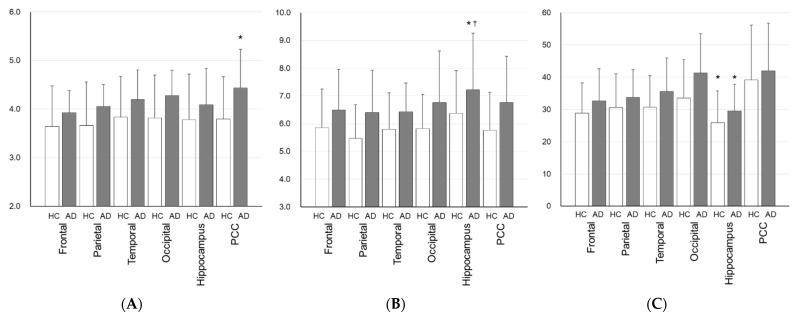
Mean values of regional K_in_ (mL/min/100 g) (**A**), k_3_ (×10^2^ /min) (**B**), and K_1_ (mL/min/100 g) (**C**) obtained by the manual ROIs method. K_in_ of eAD (gray) in the PCC region was significantly greater than those of the frontal and parietal regions in HCs (white) (* *p* < 0.05). On the other hand, k_3_ of eAD in the hippocampus was greater than other cortical values of HCs (* *p* < 0.05), particularly in the parietal region (^†^
*p* < 0.01). K1 in the hippocampus showed significant decreases for both groups (^†^
*p* < 0.01) compared with the frontal and parietal regions including PCC. SUV did not show regional differences (see Table 2).

**Figure 6 antioxidants-11-01022-f006:**
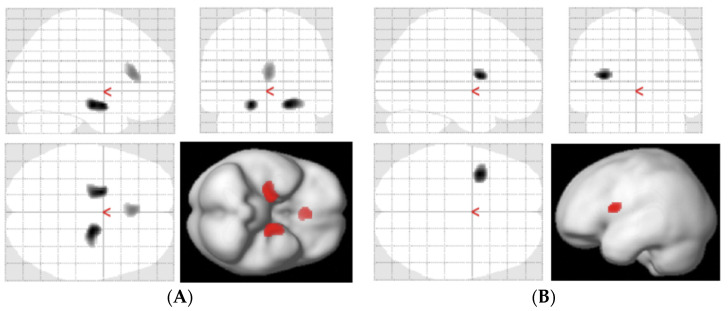
SPM analysis showed regional differences between eAD and HCs. (**A**) HC minus eAD (SUV) showed regional reductions of eAD in ACC and bilateral hippocampus. (**B**) eAD minus HC (K_in_) showed a regional increase in left central operculum of eAD.

**Table 1 antioxidants-11-01022-t001:** Demographic characteristics of study participants (mean ± SD).

	HC	eAD
N	10	10
Age (years)	72.6 ± 9.0	71.8 ± 9.3
Sex (male/female)	5/5	7/3
Education (years)	14.0 ± 3.0	12.0 ± 1.9
Duration (years)	-	2.1 ± 1.6
MMSE score	29.2 ± 0.9	23.7 ± 2.6 *
CDR score	0.0 ± 0.0	1.0 ± 0.5 *
CDR-SB score	0.0 ± 0.0	4.7 ± 2.5 *

HC—healthy control; eAD—patients with early Alzheimer’s disease; MMSE—Mini-Mental State Examination; CDR—Clinical Dementia Rating; CDR-SB—CDR sum of boxes. * *p* < 0.0001.

**Table 2 antioxidants-11-01022-t002:** Regional parametric values of SUV, K_in_, and k_3_ in each group (mean ± SD).

SUV	HC		eAD	
Frontal	6.60 ± 0.59		7.11 ± 0.58	
Parietal	6.74 ± 0.68		7.29 ± 0.62	
Temporal	6.93 ± 0.66		7.46 ± 0.61	
Occipital	6.78 ± 0.70		7.40 ± 0.62	
Hippocampus	6.67 ± 0.59		6.96 ± 0.68	
PCC	7.26 ± 0.59		7.93 ± 0.53	
**K_in_** (×10^2^ mL/min/g)				
Frontal	3.65 ± 0.83		3.93 ± 0.45	
Parietal	3.67 ± 0.89		4.06 ± 0.45	
Temporal	3.84 ± 0.84		4.19 ± 0.61	
Occipital	3.82 ± 0.88		4.28 ± 0.52	
Hippocampus	3.78 ± 0.94		4.09 ± 0.75	
PCC	3.80 ± 0.87		4.43 ± 0.80 ^†^	
k3 (×102 /min)				
Frontal	5.86 ± 1.40		6.50 ± 1.45	
Parietal	5.47 ± 1.22		6.41 ± 1.52	
Temporal	5.79 ± 1.32		6.43 ± 1.04	
Occipital	5.83 ± 1.23		6.76 ± 1.87	
Hippocampus	6.37 ± 1.54		7.22 ± 2.05 *	
PCCk	5.76 ± 1.38		6.77 ± 1.67	

HC—healthy control; eAD—patients with early Alzheimer’s disease; SUV—standardized uptake value; PCC—posterior cingulate cortex. ^†^
*p* < 0.05 compared with frontal and parietal lobes of HCs. * *p* < 0.05 compared with all lobes and PCC of HCs.

## Data Availability

Data is contained within the article.

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
