# Peer review of "Cerebral Oxidative Stress in Early Alzheimer’s Disease Evaluated by 64Cu-ATSM PET/MRI: A Preliminary Study"

_antioxidants, 2022, doi:10.3390/antiox11051022_

Round 1

Reviewer 1 Report

These articles are interesting, but I would like to see a more comprehensive description of the development of the disease in the patients studied:

  1. What is the advantage of this method of determining the development of Alzheimer's disease, along with the already existing diagnostics?
  2. Ten patients with early AD (eAD, 7 males, 71.8 ± 9.3 y) and 10 age-matched HC (5 males, 72.6 ± 9.0 y) participated in this study (Table 1).  This data duplicates the data in Table 1.
  3. In Alzheimer’s disease (AD), the most common neurodegenerative disorder causing dementia, oxidative stress in the brain is also considered to be induced by reactive oxygen species (ROS) generated by the activation of microglial and glial response to amyloid aggregation and/or neurofibril tangles [12-15]. Whether the authors know (have tested) whether Cu-ATSM PET can show the development of oxidative stress in other theories of Alzheimer's disease, for example, the works where it was found that 30% of patients diagnosed with Alzheimer's disease neurofibrillary tangles consisting of hyperphosphorylated tau were not detected. (DOI: 10.1212/01.wnl.0000118212.41542.e7; DOI: 10.1097/00005072-199311000-00012; DOI: 10.1002/ana.410300206) или erythrocyte hypothesis (doi: 10.1001/jama.293.13.1653, doi: 10.1089/neu.1996.13.223). In addition, amyloid peptide formation is not specific for AD. They are formed in response to any damaging effect resulting in neuronal death and rearrangement of neuronal networks. The formation of beta-amyloid peptides after trauma in boxers has been shown (Semin Neurol. 2000; 20(2):179-85; Chronic traumatic brain injury associated with boxing B. D. Jordan), brain damage by various toxic agents, narcosis, brain diseases of other genesis, Down syndrome (Biochem Biophys Res Commun. 1984 Aug 16; 122(3):1131-5 Alzheimer's disease and Down's syndrome: sharing of a unique cerebrovascular amyloid fibril protein. G. G. Glenner, C. W. Wong) and so on.
  4. This result indicates that global and regional increases in oxidative stress are due to mitochondrial dysfunction in the brains of eAD patients, in association with pathological protein deposition: a. Why mitochondrial dysfunction specifically, there are many other enzyme and non-enzyme systems responsible for the development of oxidative stress and which can reduce Cu (I) to Cu (II). b. Was there any correlation between the global increase in accumulation of 64Cu-ATSM into the cerebral cortices of eAD and, for example, the activities of SOD, catalase, peroxidase, MAO B, and the concentration of reduced glutathione in the blood of the same subjects?

Author Response

First of all, we really appreciate the reviewer’s valuable comments and suggestions to our study and manuscript.

Answers:

  1. The aim of this study was to investigate the pathophysiologic changes in the brain of eAD, but not to develop a new diagnostic method using Cu-ATSM. If oxidative stress is turned out to be one of the major causes of eAD, a treatment method of reducing oxidative stress would be a promising option to prevent neuronal degeneration. We added this purpose in the introduction (page 2, line 62).
  2. Thank you for pointing out this duplication of patient information. The age and number of males were deleted from the text. (page 2, line 72)
  3. We thank the reviewer suggesting several valuable references relevant to this study. We revised the Introduction (page 2, line 52) and the Discussion (page 9, 2nd paragraph) as highlighted in yellow marker, and added the following suggested papers. (No. 18-20, 40-43)
  1. Tiraboschi P, Sabbagh MN, Hansen LA, Salmon DP, Merdes A, Gamst A, Masliah E, Alford M, Thal LJ, Corey-Bloom J. Alzheimer disease without neocortical neurofibrillary tangles: "a second look". Neurology. 2004; 62(7): 1141-1147. DOI: 10.1212/01.wnl.0000118212.41542.e7
  2. Hansen LA, Masliah E, Galasko D, TerryRD. Plaque-only Alzheimer disease is usually the lewy body variant, and vice versa. J Neuropathol Exp Neurol. 1993; 52(6): 648-654. DOI: 10.1097/00005072-199311000-00012:
  3. McKee AC, Kosik KS, Kowall NW. Neuritic pathology and dementia in Alzheimer's disease. Ann Neurol. 1991; 30(2): 156-165. DOI: 10.1002/ana.410300206:
  4. Rother RP, Bell L, Hillmen P, Gladwin MT. The clinical sequelae of intravascular hemolysis and extracellular plasma hemoglobin: a novel mechanism of human disease. JAMA. 2005; 293(13): 1653-1662. DOI: 10.1001/jama.293.13.1653:
  5. Regan RF and Panter S. Hemoglobin potentiates excitotoxic injury in cortical cell culture. J Neurotrauma. 1996; 13(4); 223-231. DOI: 10.1089/neu.1996.13.223:
  6. Jordan BD. Chronic traumatic brain injury associated with boxing. Semin Neurol. 2000; 20(2): 179-185.
  7. Glenner GG, Wong CW. Alzheimer's disease and Down's syndrome: sharing of a unique cerebrovascular amyloid fibril protein. Biochem Biophys Res Commun. 1984; 122(3): 1131-1135

4-a.  Q: Why mitochondrial dysfunction specifically, there are many other enzyme and non-enzyme systems responsible for the development of oxidative stress and which can reduce Cu (I) to Cu (II).

A: Thank you for this valuable comment. As the reviewer suggested, there are several causes of oxidative stress and which can reduce Cu (I) to Cu (II) other than mitochondrial dysfunction. We have revised the description of this issue in the Discussion (page 9, line 285) and added a sentence in the 2nd paragraph (page 9, line 304) to explain that mitochondrial dysfunction is the largest source of ROS.

4-b. Q: Was there any correlation between the global increase in accumulation of 64Cu-ATSM into the cerebral cortices of eAD and, for example, the activities of SOD, catalase, peroxidase, MAO B, and the concentration of reduced glutathione in the blood of the same subjects?

A: There is no clear evidence of a relationship between the global increase in 64Cu-ATSM accumulation and the pathogenesis eAD, but one of the potential reasons may be changes in the microenvironment induced by impaired clearance function of waste products in the brain. This potential reason was added in the discussion (page 9, line 307).

Reviewer 2 Report

The present manuscript, by Okazawa et al., shows the presence of oxidative stress in AD brains evaluated by 64Cu-ATSM PET/MRI, particularly in the cingulate cortex and hippocampus. Although the manuscript is interesting and well written, the “introduction” and “conclusions” sections should be expanded in order to better highlight the relevance of the study compared to previous ones. Methods and results are well described and supported by high-quality figures and tables.  Some references instead, are too old and should be updated.

Author Response

Comments: The present manuscript, by Okazawa et al., shows the presence of oxidative stress in AD brains evaluated by 64Cu-ATSM PET/MRI, particularly in the cingulate cortex and hippocampus. Although the manuscript is interesting and well written, the “introduction” and “conclusions” sections should be expanded in order to better highlight the relevance of the study compared to previous ones. Methods and results are well described and supported by high-quality figures and tables. Some references instead, are too old and should be updated.

Answer: Thank you for the reviewer’s kind and considerable comments. We revised the Introduction and the Conclusion so as to highlight the significance of this study (yellow markers in page 2 and page 10). We also reconsider the references and added several papers (ref. No. 16-20 and 38-43). Some of them were recommended by the other reviewer.

Round 2

Reviewer 1 Report

Many thanks to the authors for the answers to the comments.